# NEURAL COMPOSITIONAL RULE LEARNING FOR KNOWLEDGE GRAPH REASONING

**Kewei Cheng** *
Department of Computer Science, UCLA
vivancheng@cs.ucla.edu

**Nesreen K. Ahmed**
Intel Labs
nesreen.k.ahmed@intel.com

**Yizhou Sun**
Department of Computer Science, UCLA
yzsun@cs.ucla.edu

## ABSTRACT

Learning logical rules is critical to improving reasoning in KGs. This is due to their ability to provide logical and interpretable explanations when used for predictions, as well as their ability to generalize to other tasks, domains, and data. While recent methods have been proposed to learn logical rules, the majority of these methods are either restricted by their computational complexity and cannot handle the large search space of large-scale KGs, or show poor generalization when exposed to data outside the training set. In this paper, we propose an end-to-end neural model for learning compositional logical rules called NCRL. NCRL detects the best compositional structure of a rule body, and breaks it into small compositions in order to infer the rule head. By recurrently merging compositions in the rule body with a recurrent attention unit, NCRL finally predicts a single rule head. Experimental results show that NCRL learns high-quality rules, as well as being generalizable. Specifically, we show that NCRL is scalable, efficient, and yields state-of-the-art results for knowledge graph completion on large-scale KGs. Moreover, we test NCRL for systematic generalization by learning to reason on small-scale observed graphs and evaluating on larger unseen ones.

## 1 INTRODUCTION

Knowledge Graphs (KGs) provide a structured representation of real-world facts (Ji et al., 2021), and they are remarkably useful in various applications (Graupmann et al., 2005; Lukovnikov et al., 2017; Xiong et al., 2017; Yih et al., 2015). Since KGs are usually incomplete, KG reasoning is a crucial problem in KGs, where the goal is to infer the missing knowledge using the observed facts.

This paper investigates how to learn logical rules for KG reasoning. Learning logical rules is critical for reasoning tasks in KGs and has received recent attention. This is due to their ability to: (1) provide interpretable explanations when used for prediction, and (2) generalize to new tasks, domains, and data (Qu et al., 2020; Lu et al., 2022; Cheng et al., 2022). For example, in Fig. 1, the learned rules can be used to infer new facts related to objects that are unobserved in the training stage.

Moreover, logical rules naturally have an interesting property - called *compositionality*: where the meaning of a whole logical expression is a function of the meanings of its parts and of the way they are combined (Hupkes et al., 2020). To concretely explain compositionality, let us consider the family relationships shown in Fig. 2. In Fig. 2(a), we show that the rule (hasUncle ← hasMother ∧ hasMother ∧ hasSon) forms a composition of smaller logical expressions, which can be expressed as a hierarchy, where predicates (i.e., relations) can be combined and replaced by another single predicate. For example, predicates hasMother and hasMother can be combined and replaced by predicate hasGrandma as shown in Fig. 2(a). As such, by recursively combining predicates into a composition and reducing the composition into a single predicate, we can finally infer the rule head (i.e., hasUncle) from the rule body. While there are various possible hierarchical trees to represent such rules, not all of them are valid given the observed relations in the KG. For example, in

---

*work was done when author was an intern at Intel Labs

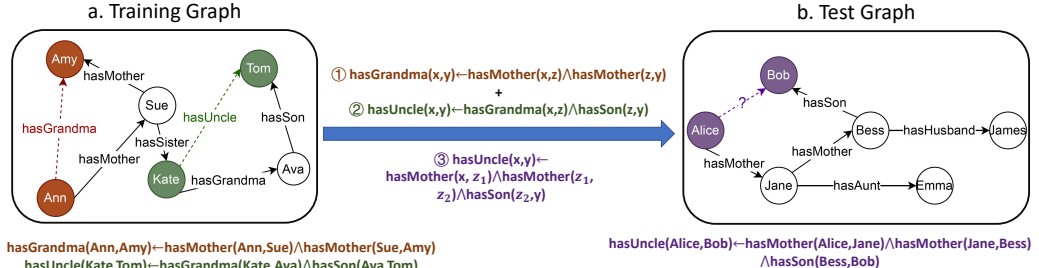

Figure 1: Illustration of how the compositionality of logical rules helps improve systematic generalization. (a) logical rule extraction from the observed graph (i.e., training stage) and (b) Inference on an unseen graph (i.e., test stage). The train and the test graphs have disjoint sets of entities. By combining logical rules ① and ② we can successfully learn rule ③ for prediction on unseen graphs.

Fig. 2(b), given a KG which only contains relations {hasMother, hasSon, hasGrandma, hasUncle}, it is possible to combine hasMother and hasSon first. However, there is no proper predicate to represent it in the KG. Therefore, learning a high-quality compositional structure for a given logical expression is critical for rule discovery, and it is the focus of our work.

In this work, our objective is to learn rules that generalize to large-scale tasks and unseen graphs. Let us consider the example in Fig. 1. From the training KG, we can extract two rules – rule ①: $\mathrm{hasGrandma}(x, y) \leftarrow \mathrm{hasMother}(x, z) \wedge \mathrm{hasMother}(z, y)$ and rule ②: $\mathrm{hasUncle}(x, y) \leftarrow \mathrm{hasGrandma}(x, z) \wedge \mathrm{hasSon}(z, y)$. We also observe that the necessary rule to infer the relation between *Alice* and *Bob* in the test KG is rule ③: $\mathrm{hasUncle}(x, y) \leftarrow \mathrm{hasMother}(x, z_1) \wedge \mathrm{hasMother}(z_1, z_2) \wedge \mathrm{hasSon}(z_2, y)$, which is not observed in the training KG. However, using compositionality to combine rules ① and ②, we can suc-

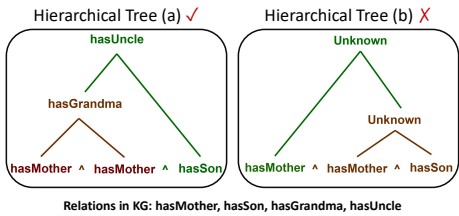

Figure 2: Learning an accurate hierarchical structure is significant for rule discovery: (a) a good compositional structure; (b) an improper compositional structure.

cessfully learn rule ③ which is necessary for inferring the relation between *Alice* and *Bob* in the test KG. The successful prediction in the test KG shows the model's ability for systematic generalization, i.e., learning to reason on smaller graphs and making predictions on unseen graphs (Sinha et al., 2019).

Although compositionality is crucial for learning logical rules, most of existing logical rule learning methods fail to exploit it. In traditional AI, inductive Logic Programming (ILP) (Muggleton & De Raedt, 1994; Muggleton et al., 1990) is the most representative symbolic method. Given a collection of positive examples and negative examples, an ILP system aims to learn logical rules which are able to entail all the positive examples while excluding any of the negative examples. However, it is difficult for ILP to scale beyond small rule sets due to their restricted computational complexity to handle the large search space of compositional rules. There are also some recent neural-symbolic methods that extend ILP, e.g., neural logic programming methods (Yang et al., 2017; Sadeghian et al., 2019) and principled probabilistic methods (Qu et al., 2020). Neural logic programming simultaneously learns logical rules and their weights in a differentiable way. Alternatively, principled probabilistic methods separate rule generation and rule weight learning by introducing a rule generator and a reasoning predictor. However, most of these approaches are particularly designed for the KG completion task. Moreover, since they require an enumeration of rules given a maximum rule length $T$, the complexity of these methods grows exponentially as max rule length increases, which severely limits their systematic generalization capability. To overcome these issues, several works such as conditional theorem provers (CTPs) (Minervini et al., 2020), recurrent relational reasoning (R5) (Lu et al., 2022) focused on the model's systematicity instead. CTPs learn an adaptive strategy for selecting subsets of rules to consider at each step of the reasoning via gradient-based optimization while R5 performs rule extraction and logical reasoning with deep reinforcement learning equipped with a dynamic rule memory. Despite their strong generalizability to larger unseen graphs beyond the training sets (Sinha et al., 2019), they cannot handle KG completion tasks for large-scale KGs due to their high computational complexity.

In this paper, we propose an end-to-end neural model to learn compositional logical rules for KG reasoning. Our proposed NCRL approach is scalable and yields state-of-the-art (SOTA) results for KG completion on large-scale KGs. NCRL shows strong systematic generalization when tested on larger unseen graphs beyond the training sets. NCRL views a logical rule as a composition of predicates and learns a hierarchical tree to express the rule composition. More specifically, NCRL breaks the rule body into small atomic compositions in order to infer the rule head. By recurrently merging compositions in the rule body with a recurrent attention unit, NCRL finally predicts a single rule head. The main contributions of this paper are summarized as follows:

- We formulate the rule learning problem from a new perspective and define the score of a logical rule based on the semantic consistency between rule body and rule head.
- NCRL presents an end-to-end neural approach to exploit the compositionality of a logical rule in a recursive way to improve the models' systematic generalizability.
- NCRL is scalable and yields SOTA results for KG completion on large-scale KGs and demonstrates strong systematic generalization to larger unseen graphs beyond training sets.

## 2 NOTATION & PROBLEM DEFINITION

**Knowledge Graph.** A KG, denoted by $\mathcal{G} = \{E, R, O\}$, consists of a set of entities $E$, a set of relations $R$ and a set of observed facts $O$. Each fact in $O$ is represented by a triple $(e_i, r_k, e_j)$.

**Horn Rule.** Horn rules, as a special case of first-order logical rules, are composed of a body of conjunctive predicates (i.e., relations are called also predicates) and a single-head predicate. In this paper, we are interested in mining chain-like **compositional Horn rules** [1] in the following form.

$$s(r_h, \mathbf{r_b}) : r_h(x, y) \leftarrow r_{b_1}(x, z_1) \wedge \cdots \wedge r_{b_n}(z_{n-1}, y) \tag{1}$$

where $s(r_h, \mathbf{r_b}) \in [0, 1]$ is the confidence score associated with the rule , and $r_h(x, y)$ is called **rule head** and $r_{b_1}(x, z_1) \wedge \cdots \wedge r_{b_n}(z_{n-1}, y)$ is called **rule body**. Combining rule head and rule body, we denote a Horn rule as $(r_h, \mathbf{r_b})$ where $\mathbf{r_b} = [r_{b_1}, \ldots, r_{b_n}]$.

**Logical Rule Learning.** Logical rule learning aims to learn a confidence score $s(r_h, \mathbf{r_b})$ for each rule $(r_h, \mathbf{r_b})$ in **rule space** to measure its plausibility. During rule extraction, the top $k$ rules with the highest scores will be selected as the learned rules.

## 3 NEURAL COMPOSITIONAL RULE LEARNING (NCRL)

In this section, we introduce our NCRL to learn compositional logical rules. Instead of using the frequency of *rule instances* to measure the plausibility of logical rules, we define the score of a logical rule as the probability that the rule body can be replaced by the rule head based on its semantic consistency. The semantic consistency between a rule body and a rule head means that the body implies the head with a high probability. An overview of NCRL is shown in Fig. 3. NCRL starts by sampling a set of paths from a given KG, and further splitting each path into short compositions using a sliding window. Then, NCRL uses a reasoning agent to reason over all the compositions to select one composition. NCRL uses a recurrent attention unit to transform the selected composition into a single relation represented as a weighted combination of existing relations. By recurrently merging compositions in the path, NCRL finally predicts the rule head. Algorithm 1 outlines the learning procedure of NCRL. Source code is available at `https://github.com/vivian1993/NCRL.git`.

### 3.1 LOGICAL RULE LEARNING WITH RECURRENT ATTENTION UNIT

As discussed in Section 1, while the rule body can be viewed as a sequence, it naturally exhibits a rich hierarchical structure. The semantics of the rule body is highly dependent on its hierarchical structure, which cannot be exploited by most of the existing rule learning methods. To explicitly allow our model to capture the hierarchical nature of the rule body, we need to learn how the relations in the rule body are combined as well as the principle to reduce each composition in the hierarchical tree into a single predicate.

---

[1] An instance of rule body of chain-like compositional Horn rules is corresponding to a path in KG

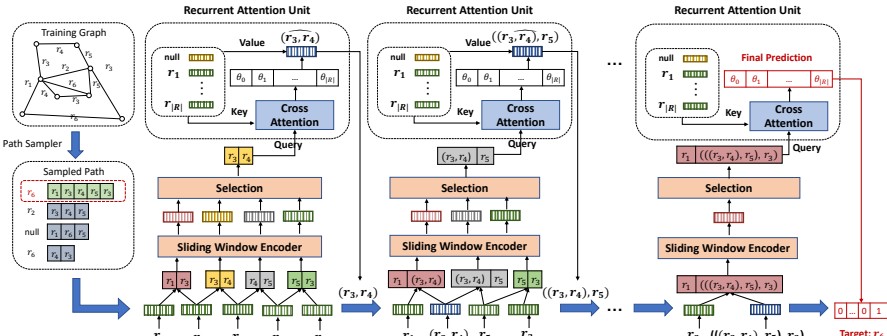

Figure 3: An overview of NCRL. It samples paths from KG (e.g.,$[r_1, r_3, r_4, r_5, r_3]$), and predicts the relations that directly connect the sampled paths (e.g.,$r_6$) based on the learned rules. NCRL takes the embeddings of predicates in the sampled paths as the input and outputs $\boldsymbol{\theta}$ as the probability of each relation to be the rule head.

### 3.1.1 HIERARCHICAL STRUCTURE LEARNING

The hierarchical structure of logical rules is learned in an iterative way. At each step, NCRL selects only one composition from the rule body and replaces the selected composition with another single predicate based on the recurrent attention unit to reduce the rule body. Although rule body is hierarchical, when operations are very local (i,e., leaf-level composition), a composition is strictly sequential. To identify a composition from a sampled path, we use a sliding window with different lengths to decompose the sampled paths into compositions of different sizes. In our implementation, we vary the size of the sliding window among $\{2, 3\}$. Given a fixed window size $s$, sliding windows are generated by a size $s$ window which slides through the rule body $\mathbf{r_b} = [r_{b_1}, \ldots, r_{b_n}]$.

**Sliding Window Encoder.** When operations are over a local sliding window (i.e., composition), the relations within a sliding window should strictly follow a chain structure. Sequence models can be utilized to encode a sliding window. Considering the tradeoff between model complexity and performance, we chose RNN (Schuster & Paliwal, 1997) over other sequence models to encode the sequence. For example, taking $i$-th sliding window whose size is 2 (i.e., $w_i = [r_{b_i}, r_{b_{i+1}}]$) as the input, RNN outputs:

$$[\mathbf{h}_i, \mathbf{h}_{i+1}] = \text{RNN}(w_i) \tag{2}$$

where $\mathbf{h}_i \in \mathbb{R}^d$ is a hidden-state of predicate $r_{b_i}$ in $w_i$. Since the final hidden-state $\mathbf{h}_{i+1}$ is usually used to represent the whole sequence, we represent the sliding window as $\mathbf{w}_i = \mathbf{h}_{i+1}$.

**Composition Selection.** $\mathbf{w}_i$ is useful to estimate how likely the relations in $i$-th window appear together. If these relations always appear together, they have a higher probability to form a meaningful composition. To incorporate this observation into our model, we select the sliding window by computing:

$$\boldsymbol{\mu} = \text{softmax}([f(\mathbf{w}_1), f(\mathbf{w}_2), \ldots, f(\mathbf{w}_{n+1-s})]) \tag{3}$$

where $f$ is a fully connected neural network. It learns the probability of $i$-th window to be a meaningful composition from its representation $\mathbf{w}_i$. $w_i$ with the highest $\boldsymbol{\mu}_i$ will be selected as the input to the recurrent attention unit.

### 3.1.2 RECURRENT ATTENTION UNIT

Note that rule induction following its underlying hierarchical structure is a recurrent process. Therefore, we propose a novel recurrent attention unit to recurrently reduce the selected composition into a single predicate until it outputs a final relation.

**Attention-based Induction.** The goal of a recurrent attention unit is to reduce the selected composition into a single predicate, which can be modeled as matching the composition with another single predicate based on its semantic consistency. Since attention mechanisms yield impressive results in Transformer models by capturing the semantic correlation between every pair of tokens in natural language sentence (Vaswani et al., 2017), we propose to utilize attention to reduce the selected composition $\mathbf{w}_i$. Note that we may not always find an existing relation to replace the selected composition. For example, given the composition $[\text{hasBrother}, \text{hasWife}]$, none of the existing relations can be used to represent it. As such, in order to accommodate unseen relations, we incorporate

a "null" predicate into potential rule heads and denote it as $r_0$. The embedding corresponding to $\mathbf{r_0}$ is set as the representation of the selected composition $\mathbf{w}_i$. In this way, when there is no direct link closing a sampled path (which means we do not have the ground truth about the rule head), we use the representation of the selected composition to represent itself rather than replace it with an existing relation. Let $H \in \mathbb{R}^{(|R|+1) \times d}$ be the matrix of the concatenation of all head relations, where $H_0 = \mathbf{w}_i \in \mathbb{R}^d$ is set as the selected composition. By taking $\mathbf{w}_i$ as a query and $H$ as the content, the scaled dot-product attention $\boldsymbol{\theta}$ [2] can be computed to estimate the semantic consistency between the selected composition and its potential heads:

$$\boldsymbol{\theta} = \text{softmax}(\frac{\mathbf{w}_i W_Q (H W_K)^T}{\sqrt{d}}) \tag{4}$$

where $W_Q, W_K \in \mathbb{R}^{d \times d}$ are learnable parameters that project the inputs into the space of query and key. $\boldsymbol{\theta} \in \mathbb{R}^{|R|+1}$ is the learned attention, in which $\boldsymbol{\theta}_j$ measures $p(r_j|w_i)$ - the probability that the selected composition can be replaced by the predicate $r_j$ based on their semantic consistency. Given $\boldsymbol{\theta}$, we are able to compute a new representation for the selected composition as a weighted combination of all head relations (i.e., values) each weighted by its attention weight:

$$\widehat{\mathbf{w}_i} = \boldsymbol{\theta} H W_V \tag{5}$$

where $\widehat{\mathbf{w}_i} \in \mathbb{R}^d$ is the new representation of the selected composition. We project the key and value to the same space by requiring $W_V = W_K$ because the keys and the values are both embeddings of relations in KG. As shown in Fig. 3, we can reduce the long rule body $[r_{b_1}, r_{b_2} \ldots, r_{b_n}]$ by recursively applying the attention unit to replace its composition $(r_{b_i}, r_{b_{i+1}})$ with a single predicate. In the final step of the prediction, the attention $\boldsymbol{\theta}$ computed following Eq. 4 collects the probability that the rule body can be replaced by each of the head relations.

## 3.2 Training and Rule Extraction

NCRL is trained in an end-to-end fashion. It starts by sampling paths from an input KG and predicts the relation which directly closes the sampled paths based on learned rules.

**Path Sampling.** We utilize a random walk (Spitzer, 2013) sampler to sample paths that connect two entities from the KG. Formally, given a source entity $x_0$, we simulate a random walk of max length $n$. Let $x_i$ denote the $i$-th node in the walk, which is generated by the following distribution:

$$p(x_i = e_i | x_{i-1} = e_j) = \begin{cases} \frac{1}{|\mathcal{N}(e_j)|}, & \text{if } (e_i, e_j) \in E \\ 0, & \text{otherwise} \end{cases} \tag{6}$$

where $|\mathcal{N}(e_j)|$ is the neighborhood size of entity $e_j$. Different from a random walk, each time after we sample the next entity $x_i$, we add all the edges which can directly connect $x_0$ and $x_i$. We denote the path connecting two nodes $x_0$ and $x_n$ as $p$, where $p = [r_{b_1}, \ldots, r_{b_n}]$, indicating $x_0 \xrightarrow{r_1} \ldots \xrightarrow{r_n} x_n$. We also denote the relation that directly connects $x_0$ and $x_n$ as $r_h$. If none of the relations directly connects the $x_0$ and $x_n$, we set $r_h$ as "null". We control the ratio of non-closed paths to ensure a majority of sampled paths are associated with an observed head relation.

**Objective Function.** Our goal is to maximize the likelihood of the observed relation $r_h$, which directly closes the sampled path $p$. The attention $\boldsymbol{\theta}$ collects the predicted probability for $p$ being closed by each of the head relations. We formulate the objective using the cross-entropy loss as:

$$- \sum_{(p, r_h) \in \mathcal{P}} \sum_{k=0}^{|R|} \mathbf{y}_k^{r_h} \log \boldsymbol{\theta}_k^p \tag{7}$$

where $\mathcal{P}$ denotes a set of sampled paths from a given KG, $\mathbf{y}^{r_h} \in \{0, 1\}^{|R|+1}$ is the one-hot encoded vector such that only the $r_h$-th entry is 1, and $\boldsymbol{\theta}^p \in \mathbb{R}^{|R|+1}$ is the learned attention for the sampled path $p$. In particular, $\boldsymbol{\theta}_0^p$ represents the probability that the sampled path cannot be closed by any existing relations in KG.

---

[2] $\boldsymbol{\theta}$ is specific to the query composition $\mathbf{w}_i$.

**Rule Extraction.** To recover logical rules, we calculate the score $s(r_h, \mathbf{r_b})$ for each rule $(r_h, \mathbf{r_b})$ in rule space based on the learned model. Given a candidate rule $(r_h, \mathbf{r_b})$, we reduce the rule body $\mathbf{r_b}$ into a single head $r_h$ by recursively merge compositions in path $\mathbf{r_b}$. At the final step of the prediction, we learn the attention $\boldsymbol{\theta} = [\boldsymbol{\theta}_0, \ldots, \boldsymbol{\theta}_{|R|}]$, where $\boldsymbol{\theta}_k$ is the score of rule $(r_k, \mathbf{r_b})$. The top $k$ rules with the highest score will be selected as learned rules.

---

**Algorithm 1:** Learning Algorithm

---
**Input:** Observed triples in KG $O$
**Output:** Relation embeddings
1  $\mathcal{P} = \text{SamplePaths}(O)$
2  **for** $(p, r_h) \in \mathcal{P}$ **do**
3      **while** $len(p) > s$ **do**
4          // Decompose $p$ with a sliding window, whose size is $s$
5          $[w_1, \ldots, w_{n+1-s}] = \text{Decompose}(p)$
6          // Select a composition
7          $[\mathbf{w}_1, \ldots, \mathbf{w}_{n+1-s}] = \text{RNN}([w_1, \ldots, w_{n+1-s}])$
8          $\mathbf{w}_i = \text{Select}([\mathbf{w}_1, \ldots, \mathbf{w}_{n+1-s}])$
9          // Apply recurrent attention unit
10         $\widehat{\mathbf{w}_i} = \text{Attn}(\mathbf{w}_i)$
11         // Reduce the sampled path $p$
12         $p = [\mathbf{r_{b_1}}, \ldots, \mathbf{w}_i, \ldots, \mathbf{r_{b_n}}]$
13     **end**
14     // Final prediction
15     $\mathbf{w} = \text{RNN}(p)$
16     Take $\mathbf{w}$ as the query and compute $\boldsymbol{\theta}$ based on Eq. 4
17     Minimize the loss in Eq. 7
18 **end**

---

## 4 EXPERIMENTS

logical rules are valuable for various downstream tasks, such as (1) *KG completion task*, which aims to infer the missing entity given the query $(h, r, ?)$ or $(?, r, t)$; (2) A more challenging *inductive relational reasoning task*, which tests the systematic generalization capability of the model by inferring the missing relation between two entities (i.e., $(h, ?, t)$) with more hops than the training data. A majority of existing methods can handle only one of these two tasks (e.g., RNNLogic is designed for the KG completion task while R5 is designed for the inductive relational reasoning task). In this section, we show that our method is superior to existing SOTA algorithms on both tasks. In addition, we also empirically assess the interpretability of the learned rules.

### 4.1 KNOWLEDGE GRAPH COMPLETION

KG completion is a classic task widely used by logical rule learning methods such as Neural-LP (Yang et al., 2017), DRUM (Sadeghian et al., 2019) and RNNLogic (Qu et al., 2020) to evaluate the quality of learned rules. An existing algorithm called forward chaining (Salvat & Mugnier, 1996) can be used to predict missing facts from logical rules.

**Datasets.** We use six widely used benchmark datasets to evaluate our NCRL in comparison to SOTA methods from knowledge graph embedding and rule learning methods. Specifically, we use the Family (Hinton et al., 1986), UMLS (Kok & Domingos, 2007), Kinship (Kok & Domingos, 2007), WN18RR (Dettmers et al., 2018), FB15K-237 (Toutanova & Chen, 2015), YAGO3-10 (Suchanek et al., 2007) datasets. The statistics of the datasets are given in Appendix A.3.1.

**Evaluation Metrics.** We mask the head or tail entity of each test triple and require each method to predict the masked entity. During the evaluation, we use the filtered setting (Bordes et al., 2013) and three evaluation metrics, i.e., Hit@1, Hit@10, and MRR.

**Comparing with Other Methods.** We evaluate our method against SOTA methods, including (1) traditional KG embedding (KGE) methods (e.g., TransE (Bordes et al., 2013), DistMult (Yang et al., 2014), ConvE (Dettmers et al., 2018), ComplEx (Trouillon et al., 2016) and RotatE (Sun et al., 2019)); (2) logical rule learning methods (e.g., Neural-LP (Yang et al., 2017), DRUM (Sadeghian et al., 2019), RNNLogic (Qu et al., 2020) and RLogic (Cheng et al., 2022)). The systematic generalizable methods (e.g., CTPs and R5) cannot handle KG completion tasks due to their high complexity.

**Results.** The comparison results are presented in Table 1. We observe that: (1) Although NCRL is not designed for KG completion task, compared with KGE models, it achieves comparable results on all datasets, especially on Family, UMLS, and WN18RR datasets; (2) NCRL consistently outperforms all other rule learning methods with significant performance gain in most cases.

### 4.1.1 ABLATION STUDY

**Performance w.r.t. Data Sparsity.** We construct sparse KG by randomly removing $a$ triples from the original dataset. Following this approach, we vary the sparsity ratio $a$ among $\{0.33, 0.66, 1\}$ and report performance on different methods over the KG completion task on Kinship dataset. As presented in Fig. 4, the performance of NCRL does not vary a lot with different sparsity ratio $a$, which is appealing in practice. More analysis of other datasets is given in Appendix A.3.2.

Table 1: KG completion. The red numbers represent the best performances among all methods, while the blue numbers represent the best performances among all *rule learning* methods.

| Methods | Models | Family | | | Kinship | | | UMLS | | |
|---|---|---|---|---|---|---|---|---|---|---|
| | | MRR | Hit@1 | Hit@10 | MRR | Hit@1 | Hit@10 | MRR | Hit@1 | Hit@10 |
| KGE | TransE | 0.45 | 22.1 | 87.4 | 0.31 | 0.9 | 84.1 | 0.69 | 52.3 | 89.7 |
| | DistMult | 0.54 | 36.0 | 88.5 | 0.35 | 18.9 | 75.5 | 0.391 | 25.6 | 66.9 |
| | ComplEx | 0.81 | 72.7 | 94.6 | 0.42 | 24.2 | 81.2 | 0.41 | 27.3 | 70.0 |
| | RotatE | 0.86 | 78.7 | 93.3 | **0.65** | **50.4** | **93.2** | 0.74 | 63.6 | 93.9 |
| Rule Learning | Neural-LP | 0.88 | 80.1 | 98.5 | 0.30 | 16.7 | 59.6 | 0.48 | 33.2 | 77.5 |
| | DRUM | 0.89 | 82.6 | 99.2 | 0.33 | 18.2 | 67.5 | 0.55 | 35.8 | 85.4 |
| | RNNLogic | 0.86 | 79.2 | 95.7 | **0.64** | **49.5** | 92.4 | 0.75 | 63.0 | 92.4 |
| | RLogic | 0.88 | 81.3 | 97.2 | 0.58 | 43.4 | 87.2 | 0.71 | 56.6 | 93.2 |
| | NCRL | **0.91** | **85.2** | **99.3** | **0.64** | 49.0 | **92.9** | **0.78** | **65.9** | **95.1** |

| Methods | Models | WN18RR | | | FB15K-237 | | | YAGO3-10 | | |
|---|---|---|---|---|---|---|---|---|---|---|
| | | MRR | Hit@1 | Hit@10 | MRR | Hit@1 | Hit@10 | MRR | Hit@1 | Hit@10 |
| KGE | TransE | 0.23 | 2.2 | 52.4 | 0.29 | 18.9 | 46.5 | 0.36 | 25.1 | 58.0 |
| | DistMult | 0.42 | 38.2 | 50.7 | 0.22 | 13.6 | 38.8 | 0.34 | 24.3 | 53.3 |
| | ConvE | 0.43 | 40.1 | 52.5 | **0.32** | 21.6 | 50.1 | 0.44 | 35.5 | 61.6 |
| | ComplEx | 0.44 | 41.0 | 51.2 | 0.24 | 15.8 | 42.8 | 0.34 | 24.8 | 54.9 |
| | RotatE | 0.47 | 42.9 | 55.7 | **0.32** | **22.8** | **52.1** | **0.49** | **40.2** | **67.0** |
| Rule Learning | Neural-LP | 0.38 | 36.8 | 40.8 | 0.24 | 17.3 | 36.2 | - | - | - |
| | DRUM | 0.38 | 36.9 | 41.0 | 0.23 | 17.4 | 36.4 | - | - | - |
| | RNNLogic | 0.46 | 41.4 | 53.1 | 0.29 | 20.8 | 44.5 | - | - | - |
| | RLogic | 0.47 | 44.3 | 53.7 | **0.31** | 20.3 | **50.1** | 0.36 | 25.2 | 50.4 |
| | NCRL | **0.67** | **56.3** | **85.0** | 0.30 | **20.9** | 47.3 | **0.38** | **27.4** | **53.6** |

† Neural-LP, DRUM, and RNNLogic exceed the memory capacity of our machines on YAGO3-10 dataset

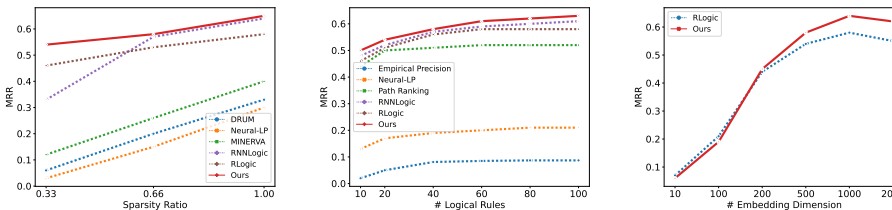

Figure 4: Performance of KG completion vs sparsity ratio on Kinship.

Figure 5: Performance of KG completion vs # logical rules on Kinship.

Figure 6: Performance of KG completion vs embedding dimension on Kinship.

**KG completion performance w.r.t. the Number of Learned Rules.** We generate $k$ rules with the highest qualities per query relation and use them to predict missing links. We vary $k$ among $\{10, 20, 40, 60, 80, 100\}$. The results on Kinship are given in Fig. 5. We observed that even with only 10 rules per relation, NCRL still givens competitive results. More analysis of other datasets is given in Appendix A.3.2.

**Performance w.r.t. Embedding Dimension.** We vary the dimension of relation embeddings among $\{10, 100, 200, 500, 1000, 2000\}$ and present the results on Kinship in Fig. 6, comparing against RLogic (Cheng et al., 2022). We see that the embedding dimension has a significant impact on KG completion performance. The best performance is achieved at $d = 1000$.

## 4.2 TRAINING EFFICIENCY

To demonstrate the scalability of NCRL, we give the training time of NCRL against other logical rule learning methods on three benchmark datasets in Table 2. We observe that: (1) Neural-LP and DRUM do not perform well in terms of efficiency as they apply a sequence of large matrix multiplications for logic reasoning. They cannot handle YAGO3-10 dataset due to the memory issue; (2)

Table 2: Training time (s) of rule learning methods

| | NeuralLP | DRUM | RNNLogic | NCRL |
|---|---|---|---|---|
| WN18RR | 1,308 | 1,146 | 1,044 | **77** |
| FB15k-237 | 23,708 | 22,428 | - | **410** |
| YAGO3-10 | - | - | - | **190** |

It is also challenging for RNNLogic to scale to large-scale KGs as it relies on all ground rules to evaluate the generated rules in each iteration. It is difficult for it to handle KG with hundreds of relations (e.g., FB15K-237) nor KG with million entities (e.g., YAGO3-10); (3) our NCRL is on average 100x faster than state-of-the-art baseline methods.

Table 3: Results of inductive relational reasoning on CLUTRR dataset. Trained on path samples with hops $\{2, 3, 4\}$ and evaluated on path samples with hops $\{5, \ldots, 10\}$. The red numbers represent the best performances while the brown numbers represent the second best performances.

| # Hops
Model | 5 Hops | 6 Hops | 7 Hops | 8 Hops | 9 Hops | 10 Hops |
|---|---|---|---|---|---|---|
| RNN | 0.93±0.06 | 0.87±0.07 | 0.79±0.11 | 0.73±0.12 | 0.65±0.16 | 0.64±0.16 |
| LSTM | 0.98±0.03 | 0.95±0.04 | 0.89±0.10 | 0.84±0.07 | 0.77±0.11 | 0.78±0.11 |
| GRU | 0.95±0.04 | 0.94±0.03 | 0.87±0.8 | 0.81±0.13 | 0.74±0.15 | 0.75±0.15 |
| Transformer | 0.88±0.03 | 0.83±0.05 | 0.76±0.04 | 0.72±0.04 | 0.74±0.05 | 0.70±0.03 |
| GNTP | 0.68±0.28 | 0.63±0.34 | 0.62±0.31 | 0.59±0.32 | 0.57±0.34 | 0.52±0.32 |
| GAT | 0.99±0.00 | 0.85±0.04 | 0.80±0.03 | 0.71±0.03 | 0.70±0.03 | 0.68±0.02 |
| GCN | 0.94±0.03 | 0.79±0.02 | 0.61±0.03 | 0.53±0.04 | 0.53±0.04 | 0.41±0.04 |
| $CTP_L$ | 0.99±0.02 | 0.98±0.04 | 0.97±0.04 | 0.98±0.03 | 0.97±0.04 | 0.95±0.04 |
| $CTP_A$ | 0.99±0.04 | 0.99±0.03 | 0.97±0.03 | 0.95±0.06 | 0.93±0.07 | 0.91±0.05 |
| $CTP_M$ | 0.98±0.04 | 0.97±0.06 | 0.95±0.06 | 0.94±0.08 | 0.93±0.08 | 0.90±0.09 |
| RLogic | 0.99±0.02 | 0.98±0.02 | 0.97±0.04 | 0.97±0.03 | 0.94±0.06 | 0.94±0.07 |
| R5 | 0.99±0.02 | 0.99±0.04 | 0.99±0.03 | 1.0±0.02 | 0.99±0.02 | 0.98±0.03 |
| NCRL | 1.0±0.01 | 0.99±0.01 | 0.98±0.02 | 0.98±0.03 | 0.98±0.03 | 0.97±0.02 |

Table 4: Results of inductive relational reasoning on GraphLog datasets for robustness analysis.

| | CTP | | RLogic | | R5 | | NCRL | |
|---|---|---|---|---|---|---|---|---|
| | ACC | Recall | ACC | Recall | ACC | Recall | ACC | Recall |
| **World 2** | 0.685±0.03 | 0.80±0.05 | 0.726±0.02 | 0.95±0.00 | 0.755±0.02 | 1.0±0.00 | **0.774±0.01** | 1.0±0.00 |
| **World 3** | 0.624±0.02 | 0.85±0.00 | 0.737±0.02 | 1.0±0.00 | 0.791±0.03 | 1.0±0.00 | **0.797±0.02** | 1.0±0.00 |
| **World 6** | 0.533±0.03 | 0.85±0.00 | 0.638±0.03 | 0.90±0.00 | 0.687±0.05 | 0.9±0.00 | **0.702±0.02** | 0.95±0.00 |
| **World 8** | 0.545±0.02 | 0.70±0.00 | 0.605±0.02 | 0.90±0.00 | 0.671±0.03 | 0.95±0.00 | **0.687±0.02** | 0.95±0.00 |

## 4.3 SYSTEMATIC GENERALIZATION

We test NCRL for systematic generalization to demonstrate the ability of NCRL to perform reasoning over graphs with more hops than the training data, where the model is trained on smaller graphs and tested on larger unseen ones. The goal of this experiment is to infer the relation between node pair queries. We use two benchmark datasets: (1) CLUTRR (Sinha et al., 2019) which is a dataset for inductive relational reasoning over family relations, and (2) GraphLog (Sinha et al., 2020) is a benchmark suite for rule induction and it consists of logical worlds and each world contains graphs generated under a different set of rules. Note that most existing rule learning methods lack systematic generalization. CTPs (Minervini et al., 2020), R5 (Lu et al., 2022), and RLogic (Cheng et al., 2022) are the only comparable rule learning methods for this task. The detailed statistics and the description of the datasets are summarized in Appendix A.4.1.

**Systematic Generalization on CLUTRR.** Table 3 shows the results of NCRL against SOTA algorithms. Detailed information about the SOTA algorithms is given in Appendix A.4.2. We observe that the performances of sequential models and embedding-based models drop severely when the path length grows longer while NCRL still predicts successfully on longer paths without significant performance degradation. Compared with systematic generalizable rule learning methods, NCRL has better generalization capability than CTPs especially when the paths grow longer. Even though R5 gives invincible results over CLUTRR dataset, NCRL shows comparable performance.
**Systematic Generalization on GraphLog.** Table 4 shows the results on 4 selected worlds. We observed that NCRL consistently outperforms other rule-learning baselines over all 4 worlds.

## 4.4 CASE STUDY OF GENERATED LOGICAL RULES

Finally, we show a case study of logical rules that are generated by NCRL on the YAGO3-10 dataset in Table 5. We can see that these logical rules are meaningful and diverse. Two rules with different lengths are presented for each head predicate. We highlight the composition and predicate which share the same semantic meaning with boldface.

## 5 RELATED WORK

**Inductive Logic Programming.** Mining Horn clauses have been extensively studied in the Inductive Logic Programming (ILP) (Muggleton & De Raedt, 1994; Muggleton et al., 1990; Muggleton, 1992; Nienhuys-Cheng & De Wolf, 1997; Quinlan, 1990; Tsunoyama et al., 2008; Zelle & Mooney, 1993). Given a set of positive examples and a set of negative examples, an ILP system learns logical

Table 5: Top rules learned on YAGO3-10. We highlight the composition and predicate which share the same semantic meaning with boldface.

| |
|---|
| $\text{isLocatedIn}(x, y) \leftarrow \textbf{isLocatedIn}(x, z) \wedge \text{isLocatedIn}(z, y)$ |
| $\text{isLocatedIn}(x, y) \leftarrow \textbf{hasAcademicAdvisor}(x, z_1) \wedge \textbf{isLocatedIn}(z_1, z_2) \wedge \text{isLocatedIn}(z_2, y)$ |
| $\text{isAffiliatedTo}(x, y) \leftarrow \text{isKnownFor}(x, z) \wedge \textbf{isAffiliatedTo}(z, y)$ |
| $\text{isAffiliatedTo}(x, y) \leftarrow \text{isKnownFor}(x, z_1) \wedge \textbf{isAffiliatedTo}(z_1, z_2) \wedge \textbf{isLeaderOf}(z_2, y)$ |
| $\text{playsFor}(x, y) \leftarrow \text{isKnownFor}(x, z) \wedge \textbf{isAffiliatedTo}(z, y)$ |
| $\text{playsFor}(x, y) \leftarrow \text{isKnownFor}(x, z_1) \wedge \textbf{playsFor}(z_1, z_2) \wedge \textbf{owns}(z_2, y)$ |
| $\text{influences}(x, y) \leftarrow \text{isPoliticianOf}(x, z) \wedge \textbf{influences}(z, y)$ |
| $\text{influences}(x, y) \leftarrow \text{isPoliticianOf}(x, z_1) \wedge \textbf{influences}(z_1, z_2) \wedge \textbf{influences}(z_2, y)$ |

rules which are able to entail all the positive examples while excluding any of the negative examples. Scalability is a central challenge for ILP methods as they involve several steps that are NP-hard. Recently, several differentiable ILP methods such as Neural Theorem Provers (Rocktäschel & Riedel, 2017; Campero et al., 2018; Glanois et al., 2022) are proposed to enable a continuous relaxation of the logical reasoning process via gradient descent. Different from our method, they require predefined hand-designed, and task-specific templates to narrow down the rule space.

**Neural-Symbolic Methods.** Very recently, several methods extend the idea of ILP by simultaneously learning logical rules and weights in a differentiable way. Most of them are based on neural logic programming. For example, Neural-LP (Yang et al., 2017) enables logical reasoning via sequences of differentiable tensor multiplication. A neural controller system based on attention is used to learn the score of a specific logic. However, Neural-LP could learn a higher score for a meaningless rule because it shares an atom with a useful rule. To address this problem, RNNs are utilized in DRUM (Sadeghian et al., 2019) to prune the potential incorrect rule bodies. In addition, Neural-LP can learn only chain-like Horn rules while NLIL (Yang & Song, 2019) extends Neural-LP to learn Horn rules in a more general form. Because neural logic programming approaches involve large matrix multiplication and simultaneously learn logical rules and their weights, which is nontrivial in terms of optimization, they cannot handle large KGs, such as YAGO3-10. To address this issue, RNNLogic (Qu et al., 2020)) is proposed to separate rule generation and rule weight learning by introducing a rule generator and a reasoning predictor respectively. Although the introduction of the rule generator reduces the search space, it is still challenging for RNNLogic to scale to KGs with hundreds of relations (e.g., FB15K-237) or millions of entities (e.g., YAGO3-10).

**Systematic Generalizable Methods.** All the above methods cannot generalize to larger graphs beyond training sets. To improve models' systematicity, Conditional Theorem Provers (CTPs) is proposed to learn an optimal rule selection strategy via gradient-based optimization. For each sub-goal, a select module produces a smaller set of rules, which is then used during the proving mechanism. However, since the length of the learned rules influences the number of parameters of the model, it limits the capability of CTPs to handle the complicated rules whose depth is large. In addition, due to its high computational complexity, CTPs cannot handle KG completion tasks for large-scale KGs. Another reinforcement learning-based method - R5 (Lu et al., 2022) is proposed to provide a recurrent relational reasoning solution to learn compositional rules. However, R5 cannot generalize to the KG completion task due to the lack of scalability. It requires pre-sampling for the paths that entail the query. Considering that all the triples in a KG share the same training graph, even a relatively small-scale KG contains a huge number of paths. Thus, it is impractical to apply R5 to even small-scale KG for rule learning. In addition, R5 employs a hard decision mechanism for merging a relation pair into a single relation, which makes it challenging to handle the widely existing uncertainty in KGs. For example, given the rule body $hasAunt(x, z) \wedge hasSister(z, y)$, both $hasMother(x, y)$ and $hasAunt(x, y)$ can be derived as the rule head. The inaccurate merging of a relation pair may result in error propagation when generalizing to longer paths. Although RLogic (Cheng et al., 2022) are generalizable across multiple tasks, including KG completion and inductive relation reasoning, they couldn't systematically handle the compositionality and outperformed by NCRL.

## 6 CONCLUSION

In this paper, we propose NCRL, an end-to-end neural model for learning compositional logical rules. NCRL treats logical rules as a hierarchical tree and breaks the rule body into small atomic compositions in order to infer the head rule. Experimental results show that NCRL is scalable, efficient, and yields SOTA results for KG completion on large-scale KGs.

## ACKNOWLEDGEMENTS

This work was partially supported by NSF 2211557, NSF 2303037, NSF 1937599, NSF 2119643, NASA, SRC, Okawa Foundation Grant, Amazon Research Awards, Amazon Fellowship, Cisco research grant, Picsart Gifts, and Snapchat Gifts.

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
