# OpenReview forum: "Neural Compositional Rule Learning for Knowledge Graph Reasoning"
_ICLR.cc/2023/Conference — ICLR 2023 poster_

### Official Review · Reviewer_WAeF · 2022-10-25

**Confidence:** 4
**Correctness:** 1
**Technical Novelty And Significance:** 2
**Empirical Novelty And Significance:** 2
**Recommendation:** 6

**Clarity, Quality, Novelty And Reproducibility:**

The quality of the paper is questionable in terms of technical soundness. The performance of the proposed approach depends on random sampling, and it is unclear how randomness impacts the performance and how the proposed approach guarantees a stable performance in considering tasks. The clarity of the paper is generally good except the aforementioned issues in technical soundness. The main originality lies in learning chain-like rules by composing relational sub-chains within a window of fixed size. A similar idea for hierarchical rule induction has been published in ICML2022 [1]. The authors may add the comparisons with [1] in Related Work to highlight their originality. The source code is provided in the supplemental material but there are no guidelines provided to download the datasets and to reproduce the experimental results.

[1] Claire Glanois, Zhaohui Jiang, Xuening Feng, Paul Weng, Matthieu Zimmer, Dong Li, Wulong Liu, Jianye Hao: Neuro-Symbolic Hierarchical Rule Induction. ICML 2022: 7583-7615


**Strength And Weaknesses:**

Strengths:

(1) Learning chain-like rules by composing the body atoms step by step has a lower time complexity than learning rules by generating all body atoms at a time. This efficiency benefit has been confirmed by experiments.

(2) Rather extensive experiments on two different tasks have been conducted to demonstrate that the proposed approach NCRL outperforms the state-of-the-arts. In particular, in the link prediction task NCRL is shown to achieve the best performance among all state-of-the-art rule learning approaches that do not consider entity embeddings.

Weaknesses:

(1) The superiority in performance depends on the sampled results on relation paths in the given knowledge graph. That is, this superiority depends on random sampling, and thus it is not stable and unconvincing. The paper should report the average and std results for different runs of the proposed approach. It would be better if different sampling strategies can also be compared with each other to draw a guideline for selecting an appropriate sampling method.

(2) The learnt rules are more restricted than ordinary chain-like Horn rules since they require that adjacent body atoms appearing in siblings of the composition hierarchy should be able to compose a named relation. It is unclear whether this lower expressivity impacts the performance in link prediction or inductive relational reasoning, and whether this restriction can be removed to increase the expressivity to meet broader application requirements.


**Summary Of The Paper:**

The paper proposes a new approach to learning chain-like Horn rules by constructing the relation composition hierarchy in rule bodies from sample paths and by extracting the rule bodies from learnt relation embeddings. Experimental results on two tasks, namely link prediction and inductive relational reasoning, demonstrate the advantages of the proposed approach.

**Summary Of The Review:**

The paper has a major problem in technical soundness by considering that it is based on random sampling and may not yield stable performance. The expressivity of learnt rules is lower than ordinary chain-like Horn rules. This means that the proposed approach is probably inferior to state-of-the-art methods that learn ordinary chain-like Horn rules, in scenarios where the expressivity of ideal rules is higher than the supported expressivity.

AFTER RESPONSE:
I increased the overall score since the authors reasonably addressed my main concerns.

---

> ### Author Response · Authors · 2022-11-19
> **Reply to Reviewer WAeF**
>
> We thank the reviewer for all the insightful comments. We now address the concerns raised by the reviewer one by one as below.
>
> Q1: The superiority in performance depends on the sampled results on relation paths in the given knowledge graph. That is, this superiority depends on random sampling, and thus it is not stable and unconvincing. The paper should report the average and std results for different runs of the proposed approach. It would be better if different sampling strategies can also be compared with each other to draw a guideline for selecting an appropriate sampling method.
>
> As suggested, we now added an ablation study to investigate the impact of our sampling approach on the model performance. We reported the results of different runs of our model on multiple datasets for KG completion under the same setting in appendix A.4.2, which is highlighted in purple. The results are given in Table 4 as well as below. We observe that our sampling strategy is stable and has a small standard deviation.
>
> $$
> \begin{array}{@{}|c|ccc|ccc|ccc|@{}}
> \hline
> &&\text{Family}& & &\text{Kinship}& & &\text{UMLS} & \\\\
> &\text{MRR} & \text{Hit@1} & \text{Hit@10} & \text{MRR} & \text{Hit@1} & \text{Hit@10} & \text{MRR} & \text{Hit@1} &  \text{Hit@10} \\\\
> \hline
> \textbf{Standard Deviation} & \textbf{0.000} & \textbf{0.330} & \textbf{0.170} & \textbf{0.005} & \textbf{0.250} & \textbf{0.294} & \textbf{0.005} & \textbf{0.245} & \textbf{ 0.125} \\\\
> \hline
> \end{array}
> $$
>
> Q2:  The learned rules are more restricted than ordinary chain-like Horn rules since they require that adjacent body atoms appearing in siblings of the composition hierarchy should be able to compose a named relation. It is unclear whether this lower expressivity impacts the performance in link prediction or inductive relational reasoning, and whether this restriction can be removed to increase the expressivity to meet broader application requirements.
>
> There seems to be a misunderstanding. We want to clarify that we do not require the composition to be able to compose a named relation. It is possible that the composition can not compose a named relation. In order to accommodate unknown relations, we incorporate a “null”' predicate into potential rule heads to broaden the expressivity. Note that “null” is used as a placeholder to denote there is no existing relation to close the loop rather than to represent all the unobserved relations with a default relation. We always use the representation of the query (the rule body) to be the representation of “null”. When the attention score on “null” is the highest, it implies the query does not lead to any existing relations. When the final prediction is “null”, it indicates we cannot find a rule with the current rule body. We added the detailed discussion in Section 3.1.2 highlighted in purple.
>
> Q3:There are no guidelines provided to download the datasets
>
> Since there are limitations on the size of the supplementary materials, we were unable to upload all datasets. We now added the datasets in the following link:
>
> https://anonymous.4open.science/r/ICLR2023_Datasets-C317
>
> Q4: Comparison to Neuro-Symbolic Hierarchical Rule Induction (HRI)
>
> We were not aware of this recently published reference, so thanks for pointing it out to us. As suggested, we have now cited this reference in our related work section.
>
> Now let’s clarify how this work relates to our work. The work by Glanois et al. (ICML’22) extends logical rule induction[1] whose representation model is inspired by Neural Theorem Provers (NTPs) to a hierarchical structure with a pre-defined set of meta-rules templates, and was shown to be able to tackle many of the classical inductive logic programming (ILP) benchmark tasks. Unlike our approach, the work by Glanois et al. (ICML’22) requires a pre-defined set of meta-rule templates, which is unclear how this will generalize to tasks beyond classical ILP and it remains as one of their major future work directions to extend the meta-rule templates to broaden expressivity to other tasks beyond ILP.
>
> On the other hand, we compare with the state-of-the-art methods that apply to our settings including Conditional Theorem Provers (CTPs) which extends NTPs to learn optimal rule selection and can handle larger-scale settings compared to NTPs.
>
> [1] Campero, A., Pareja, A., Klinger, T., Tenenbaum, J., and Riedel, S. Logical rule induction and theory learning using neural theorem proving. arXiv preprint arXiv:1809.02193, 2018.

---

### Official Review · Reviewer_N8Wt · 2022-10-26

**Confidence:** 5
**Clarity, Quality, Novelty And Reproducibility:** See comments above.
**Correctness:** 4
**Technical Novelty And Significance:** 3
**Empirical Novelty And Significance:** 3
**Recommendation:** 8

**Strength And Weaknesses:**

Strengths:
1. The problem of learning logic rules is important for knowledge graph reasoning, which could also inspire more advanced models for other reasoning problems.
2. The idea of breaking rules into small compositions is intuitive and effective, as illustrated in Figure 1. The proposed approach is easy to follow.
3. The proposed approach achieves SOTA results on many datasets such as WN18RR.

Weaknesses:
1. Compositional Horn rules have been widely used for knowledge graph reasoning. It is unclear whether the proposed framework can generalize to other more complicated rules.
2. The experiment can be further improved.

Below are some detailed comments:
1. NCRL uses compositional Horn rules for knowledge graph reasoning. Although the results are impressive, such logic rules have been widely explored. In order to develop more advanced models for knowledge graph reasoning and other reasoning tasks, I believe that some more complicated logic rules are needed (e.g., some tree-structured rules). I wonder whether NCRL can also model such logic rules.
2. In Eq. (2), the proposed model uses a RNN for representation learning. Why does NCRL choose to use RNN? Whether some more advanced sequence models such as Transformer can help?
3. On WN18RR, NCRL achieves SOTA results while on FB15K-237 the results are slightly worse. Is there any intuition why NCRL gets better results on WN18RR? Is this because WN18RR is more sparse in terms of the number of triples? This might be important if we would like to develop better models in the future.
4. In the experiment, only some small datasets are considered. Is it possible to apply NCRL to some larger datasets, such as the OGB dataset?

**Summary Of The Paper:**

This paper studies compositional rule learning for knowledge graph reasoning, and a model called the NCRL is proposed. NCRL is able to break a rule into small compositions for reasoning, and a recurrent attention unit is proposed to integrate different relational paths. Experimental results on several benchmarks are impressive.

**Summary Of The Review:**

See comments above.

---

> ### Author Response · Authors · 2022-11-19
> **Reply to Reviewer N8Wt**
>
> We thank the reviewer for all the insightful comments. We now address the concerns raised by the reviewer one by one as below.
>
> Q1: NCRL uses compositional Horn rules for knowledge graph reasoning. Although the results are impressive, such logic rules have been widely explored. In order to develop more advanced models for knowledge graph reasoning and other reasoning tasks, I believe that some more complicated logic rules are needed (e.g., some tree-structured rules). I wonder whether NCRL can also model such logic rules.
>
> Our proposed method focuses on learning compositional Horn rules. Learning tree-structured rules is a more challenging task as it involves a much more complicated rule searching space. We agree complex rules are in great need, which is our future work.
>
>
> Q2: In Eq. (2), the proposed model uses a RNN for representation learning. Why does NCRL choose to use RNN? Whether some more advanced sequence models such as Transformer can help?
>
> The subsequences extracted by the sliding windows are sequential, and thus we chose to use sequence models such as RNN to encode them. The reason we chose RNN over Transformer is due to the complexity concern. After we’ve done an ablation study as given below, we observed that the performance of the Transformer-based encoder is even worse, likely due to the overfitting issue caused by the large parameter space of the Transformer.
>
> $$
> \begin{array}{@{}|c|ccc|ccc|ccc|@{}}
> \hline
> &&\text{Family}& & &\text{Kinship}& & &\text{UMLS} & \\\\
> \text{Encoder}&\text{MRR} & \text{Hit@1} & \text{Hit@10} & \text{MRR} & \text{Hit@1} & \text{Hit@10} & \text{MRR} & \text{Hit@1} &  \text{Hit@10} \\\\
> \hline
> \text{Transformer} & 0.84 & 75.6 & 95.3 & 0.50 & 33.7 & 79.1 & 0.62 & 48.8  & 81.2 \\\\
> \text{RNN} & \textbf{0.92} & \textbf{85.6} & \textbf{99.6} & \textbf{0.65} & \textbf{49.4} & \textbf{93.6} & \textbf{0.78} & \textbf{66.1} & \textbf{95.2} \\\\
> \hline
> \end{array}
> $$
>
> Q3: On WN18RR, NCRL achieves SOTA results while on FB15K-237 the results are slightly worse. Is there any intuition why NCRL gets better results on WN18RR? Is this because WN18RR is more sparse in terms of the number of triples? This might be important if we would like to develop better models in the future.
>
> Thanks for pointing out the performance difference. We provide our conjecture below.
>
> WN18RR (WordNet) is a KG with 11 relations (e.g., synonym and antonym) on 41K words. The rule space formed by 11 relations is relatively small, and the closed paths are easy to obtain (e.g., antonym, antonym => synonym). The high-quality rules are supported by abundant closed paths, which is easier for our model to learn.
> FB15K-237 (Freebase) is a KG with 237 relations on 14.5K entities, covering much more diverse relations over a smaller set of entities compared to WN18RR. The rule space is much bigger due to the bigger size of relations. Even worse, those relations are not well connected, meaning closed paths are rare and rules are also more difficult to find. In this case, the embedding-based approaches have more advantages for KG completion tasks since they use entities.
>
> Q4: In the experiment, only some small datasets are considered. Is it possible to apply NCRL to some larger datasets, such as the OGB dataset?
>
> Thanks for your feedback.  We have results on large-scale benchmark KGs (e.g., YAGO3-10 in Table 1). YAGO3-10 contains 1,089,040 triples, 37 relations, and 123,182 entities, which has the same scale as some OGB datasets, such as collab and arxiv. Since most of OGB datasets including collab and arxiv are homogeneous, they are not applicable to our setting. We also provide the theoretical time complexity analysis for our proposed method in Appendix A.6. The training time is only related to the embedding size of the relations and the number of sampled paths. Since our model requires only schema-level paths (keep only relations by ignoring all entities along a path instance) as the input, the required number of sampled paths only depends on the total number of relations in KG instead of the number of entities or the number of triples in KGs.

---

### Official Review · Reviewer_FLaJ · 2022-10-26

**Confidence:** 4
**Correctness:** 3
**Technical Novelty And Significance:** 3
**Empirical Novelty And Significance:** 3
**Recommendation:** 8

**Clarity, Quality, Novelty And Reproducibility:**

The paper's clarity and reproducibility needs improving. The quality and novelty of the work is good.

**Strength And Weaknesses:**

Strengths:
 - Good technical novelty
 - Good empirical evaluation: demonstrating good performance and good scalability

 Weaknesses:
 - The clarity of the paper could be improved, in particular variable names are re-used and mean different things in different sections of the paper, which is an unnecessary hurdle for the reader.
 - Some questions about the implementation remain that would be necessary for reproducibility:
	- What are the hyperparameters there is no table in the appendix for this as far as I can see (for the results in Table 1 & 3)?
	- Will you publish code upon acceptance?

 - Certain design decisions are unexplained and not studied in the ablation studies:
	- Why use an RNN for the encoding of sliding windows (due to the fixed length of window) a standard neural network would do? Did you compare against this?

Questions:
 - How is semantic consistency defined in the intro of section 3?
 - Why is an RNN architecture needed for a fixed size sliding window (eq 2.) did you try a standard Neural Network? Would it be possible to do an ablation on this?
 - How the length of sampled paths determined? Is there a maximum cut-off? Are loops automatically removed? In general the section on path sampling was difficult to understand.
 - Why did you choose to project the key and values to the same space (i.e. W_V = W_K eq5)?

Minor:
 - Please use a different variable name for w_i after the RNN stage, it's unnecessarily confusing. Similarly, please rename the hyperparameter theta as theta is already used in eq4.
 - There is a typo in 4.3, 3rd line at the very end.
 - When defining H in 3.1.2 there are missing brackets in the power as multiplication binds more tightly than addition i.e. it should be (|R|+1) x d.

**Summary Of The Paper:**

The paper proposes to learn logic rules by recursively encoding subsets of paths into head relations. The paths are samples from the KG. The method to do this is using an RNN to encode each element of the sliding window on the path, selecting one window with a softmax, and using an attention mechanism to predict the summarising head entity.

The authors demonstrate the scalability advantage and the competitive performance of their method against strong rule-learning based methods on a variety of standard benchmarks.


**Summary Of The Review:**

The paper makes a solid technical contribution and validates it well empirically. Some questions remain (see above).

I will raise my score if my concerns regarding clarity and reproducibility are addressed.

EDIT: The rebuttal has addressed my concerns, I have adapted my score accordingly.

EDIT EDIT: Due to the concerns regarding the empirical evaluation, I have to lower my score again.

EDIT EDIT EDT: The new results have resolved the concerns surrounding the empirical evaluation.

---

> ### Author Response · Authors · 2022-11-18
> **Reply to Reviewer FLaJ**
>
> We thank the reviewer for all the insightful comments. We now address the concerns raised by the reviewer one by one as below.
>
> Q1: The clarity of the paper could be improved, in particular, variable names are re-used and mean different things in different sections of the paper, which is an unnecessary hurdle for the reader.
>
> Thank you for pointing this out. As suggested, we revised the manuscript to ensure $\theta$ is used specifically for the final prediction. We have revised the original manuscript and highlighted it in purple.
>
> As for the re-use of $w_i$, we would like to clarify that the regular font $w_i$ is used to represent a sub-sequence of the rule body covered by the sliding window i. The bold font  $\mathbf{w}_i$ is used to represent the embedding of the sequence$w_i$. This is following the convention in the KG embedding community.
>
> Q2 Some questions about the implementation remain that would be necessary for reproducibility:
>
> 2.1 What are the hyperparameters there is no table in the appendix for this as far as I can see (for the results in Table 1 & 3)?
>
> As suggested, we provide the hyperparameters in Table 1 in Appendix A.3 and highlighted the changes in purple.
>
> 2.2 Will you publish code upon acceptance?
>
> Yes, we will publish the code on GitHub upon acceptance. We also added the code in the supplementary materials.
>
> Q3 Why use an RNN for the encoding of sliding windows. Did you try a standard Neural Network? Would it be possible to do an ablation on this?
>
> Thanks for the feedback. We chose RNN due to the sequential nature of the subsequences extracted by the sliding windows. As suggested we did the ablation study, where we used the standard neural network as the sliding window encoder and investigated its performance in appendix A.4.2, which is highlighted in purple. The results are given in Table 6 as well as below.  We observe that in comparison to the standard neural network, RNN encoder achieves better performance.
>
> $$
> \begin{array}{@{}|c|ccc|ccc|ccc|@{}}
> \hline
> &&\text{Family}& & &\text{Kinship}& & &\text{UMLS} & \\\\
> \text{Encoder}&\text{MRR} & \text{Hit@1} & \text{Hit@10} & \text{MRR} & \text{Hit@1} & \text{Hit@10} & \text{MRR} & \text{Hit@1} &  \text{Hit@10} \\\\
> \hline
> \text{MLP} & 0.88 & 80.2 & 98.0 & 0.56 & 38.0 & 84.9 & 0.69 & 55.8  & 87.4 \\\\
> \text{RNN} & \textbf{0.92} & \textbf{85.6} & \textbf{99.6} & \textbf{0.65} & \textbf{49.4} & \textbf{93.6} & \textbf{0.78} & \textbf{66.1} & \textbf{95.2} \\\\
> \hline
> \end{array}
> $$
>
> Q4 How is semantic consistency defined in the intro of section 3?
>
> The semantic consistency between a rule body and a rule head means that the body implies the head with a high probability. Technically, it means the corresponding \theta_i has a high value.
>
> For example, a relation sequence [mother, mother]  predicts the single relation grandmother with a high probability, indicating the following logical rule holds true
>
> mother(a,b) ^mother(b,c) -> grandmother(a,c)
>
> As suggested, we added clarification in Section 3.
>
> Q5 How are the length of sampled paths determined? Is there a maximum cut-off? Are loops automatically removed? In general, the section on path sampling was difficult to understand.
>
> We set the length of sampled paths during training as 3 in our experiments. Since our model incorporates the compositionality of logical rule during learning, it is able to generalize to longer rules even when trained using short sampled paths. Therefore, the length of sampled paths can be set as a small number, which greatly improves the training efficiency.
>
> We won’t remove loops as they are preferred during sampling. Loops are cycles in the graph, which consist of two components: (1) a muti-hop path connecting two entities x_i and x_j and (2) a single relation directly connects entities x_i and x_j. Each cycle in KG is corresponding to an observed rule instance, where the muti-hop path corresponds to the rule body and the single relation corresponds to the rule head.
>
> As suggested, we revised the section on path sampling and highlighted it in purple.
>
> Q6 Why did you choose to project the key and values to the same space (i.e. W_V = W_K eq5)?
>
> We project the key and values to the same space because the keys and the values are both embedding of relations in KG. Projecting the key and values to the same space is used by many transformer-based models. For example, in translation tasks, the keys and the values are both the embeddings of the input words, and their projection matrices are shared. As suggested, we discussed these details in the revision and highlighted the changes in purple in Section 3.1.2.
>
> Q7: Typo issues.
>
> Thank you for pointing these typos out. We have fixed the typos in the revision.

---

### Official Review · Reviewer_XZA5 · 2022-10-27

**Confidence:** 5
**Correctness:** 3
**Technical Novelty And Significance:** 3
**Empirical Novelty And Significance:** 3
**Recommendation:** 6

**Clarity, Quality, Novelty And Reproducibility:**

### Novelty


I find the proposed compositional view of the chain-like rule very interesting and inspiring. I particularly like the example shown in the introduction and Fig 1. The proposed model is technically sound, although there might be some aspects needing further justification. The experiments are well-designed and support the claims. A solid contribution to the ILP community.


### Quality


Section 3.2 suggests that for every pair of entities with a sampled path, one sets the head to "null" if there is no direct connection. And with Eq 7, this effectively makes the model predict "null" for every pair of entities with missing direct edges. I'm not sure if this is correct: this essentially leads to a closed-world setting, where every possible missing edge is assigned with a specific label (i.e., null). This can be problematic with many real-world KGs that have many missing edges: for example, there might be two pairs of $(x_0,x_n)_1$ and $(x_0,x_n)_2$ with the same path $p_1 = p_2$, but $(x_0,x_n)_1$ is connected with $r_h=hasMother$ while $(x_0,x_n)_2$ is not. In this proposed setting, the model is told to predict hasMother for $(x_0,x_n)_1$ and null for $(x_0,x_n)_2$ even though they have the same path.

I find the scalability argument not very convincing. The proposed model should have similar asymptotic complexity as other backward-chaining methods such as NeuralLP, as all of them are based on the same personalized random walk mechanism. The difference is just that NeuralLP computes the state probability directly with matrix multiplication, whereas NCLR samples from it. The difference in 4.2 seems to be more of a consequence of implementational optimization. For example, by default NeuralLP multiplies the entire matrix for every query, one can readily optimize it by storing the local adjacency matrices up to the max search step.

With that being said, the path sampling phase may need more elaboration. Right now, it is unclear how the max search step is set, how many samples are collected per pair, and how the duplicates are handled. Also, it is also important to discuss how sensitive the model is w.r.t these hyperparameters.

The authors set the window size to {2,3} throughout the experiments. It would be nice to have an experiment to show how model performance changes w.r.t the window size.


### Clarity


The paper is well-written and generally easy to follow. Below are some minor issues.


In Fig 3 and section 3.1.1, what is $r_i$? Is it an embedding vector of a unique relation?


Minor issues
	- "infer the relation between nodel pair query"
	- Conjunctions missing in some rules in Table 5


**Strength And Weaknesses:**

Strength
- Proposes an interesting perspective of viewing chain-like rules in KGs via hierarchical compositions, which is different from the classical multi-hop pathfinding view.


Weaknesses
- Some design choices (see below) need more justification


**Summary Of The Paper:**

This work proposes a novel differentiable model for the ILP problem, i.e., mining logic from KGs. To do this, the authors introduce a compositional view of the chain-like rules and design a learning schema where the model learns to reduce a path into a single edge. The proposed model, namely NCRL, is an RNN-transformer hybrid model that parameterizes this schema with sequences of soft attention. In the experiments, NCRL demonstrates better systematic generalization in several benchmarks.

**Summary Of The Review:**

In summary, this paper introduces an interesting view of the chain-like rule and proposes a technically sound model that takes advantage of this observation. I do have some concerns about the methodology and technical details and hope the authors could address them, but in general, I really enjoy reading the paper. That said, I recommend acceptance.

---

> ### Author Response · Authors · 2022-11-18
> **Reply to Reviewer XZA5**
>
> We thank the reviewer for all the insightful comments. We now address the concerns raised by the reviewer one by one as below.
>
> Q1: Concerns about setting the head to "null" if there is no direct connection
>
> We’d like to clarify that “null” is used as a placeholder to denote there is no existing relation to close the loop rather than to represent all the unobserved relations with a default relation. Technically, we always use the representation of the query (the rule body) to be the representation of “null”. When the attention score on “null” is the highest, it implies the query does not lead to any existing relations.
>
> In addition, we also control the ratio of non-closed paths during path sampling to ensure a majority of sampled paths are associated with an observed head relation.
>
> As suggested, we have now revised this part of the manuscript and highlighted the revision in Section 3.1.2.
>
> Q2:  Concerns on scalability compared to backward-chaining methods
>
> Thanks for your feedback. There are two major differences between our proposed method and other backward-chaining methods such as NeuralLP: (1) Backward-chaining methods require computing full matrix multiplication in order to instantiate every possible rule in the rule space (under a length constraint). Our proposed method is a representation-learning-based approach, and only a small amount of samples are enough for us to learn the model. As suggested, we have now compared the complexity of our method and some backward-chaining methods in appendix A.6, which is highlighted in purple;  (2) As a special case of (1),  backward-chaining methods require to enumerate every possible rule given a maximum rule length T while our model can be trained on short path samples to learn longer rules.
>
> Q3:  It would be nice to have an experiment to show how model performance changes w.r.t the window size.
>
> Thank you for the suggestion. We now added an ablation study to investigate how the model performance changes w.r.t. window size in Appendix A.4.2, which is highlighted in purple. The results are given in Table 5 as well as below. We observe that we consistently achieve the best performance by setting the window size to 2. The major reason is that we apply rules with a maximum length of 3 for KG completion task. In this case, if we set the size of the sliding window as 3, our model cannot leverage the compositionality and thus results in worse performance.
>
> $$
> \begin{array}{@{}|c|ccc|ccc|ccc|@{}}
> \hline
> \text{Window}&&\text{Family}& & &\text{Kinship}& & &\text{UMLS} & \\\\
> \text{Size}&\text{MRR} & \text{Hit@1} & \text{Hit@10} & \text{MRR} & \text{Hit@1} & \text{Hit@10} & \text{MRR} & \text{Hit@1} &  \text{Hit@10} \\\\
> \hline
> 2 & \textbf{0.92} & \textbf{85.6} & \textbf{99.6} & \textbf{0.65} & \textbf{49.4} & \textbf{93.6} & \textbf{0.78} & \textbf{66.1} & \textbf{95.2} \\\\
> 3 & 0.90 & 82.3 & 99.5 & 0.60 & 43.1 & 88.9 & 0.72 & 59.9  & 89.3 \\\\
> \hline
> \end{array}
> $$
>
> Q4: In Fig 3 and section 3.1.1, what is $r_i$
>
> $r_i$ denotes the ith relation, which is associated with a  learnable embedding vector as the input to the RNN component of the model. As suggested, we have revised the manuscript to make it clear and highlighted the change in purple in Fig 3.
>
> Q5: Minor typo issues:(1) "infer the relation between nodel pair query"; (2) Conjunctions missing in some rules in Table 5.
>
> Thank you for pointing these typos out. We have fixed the typos in the revision.

---

> > ### Comment · Reviewer_XZA5 · 2022-12-03
> > **Comments**
> >
> > Thanks for the response. These justifications are sensible to me.
> >
> > However, after reading the public comment and the response, I do have a **serious concern** about the potential data leakage. First of all, using facts+train+valid+test triples during inference is uncommon for KG completion. The authors claim this is the same as that of NTPs, but NTPs was not even included as the baseline. And as far as I know, backward-chaining methods like NeuralLP only use fact triples during inference.
> >
> > Still, this setting could still be sensible if the authors have rerun all baseline methods in this new setting, but judging from Table 1, the reported scores are either the same as that reported in the original paper (NeuralLP on FB15K-237), or even lower (DRUM). It is not clear if the results are directly taken from the original work (in which case, the comparison is unfair), or obtained by running them in the proposed setting (in which case, the scores should be presumably higher than that reported in the original work, because more data were given).
> >
> > Can authors clarify this aspect?

---

> > > ### Author Response · Authors · 2022-12-06
> > > **Reply to Reviewer XZA5**
> > >
> > > Thanks for the follow-up comments. We now address your questions below.
> > >
> > > (1) Potential Data Leakage Issue.
> > >
> > > First, we clarify that when learning the rules we didn’t use test triples. The experimental setting for rule-based inference in this paper follows a leave-one-out test, meaning all the triples except the test triple will serve as facts in the KG to predict the held-out test triple by grounding the learned rules. Due to our compositional learning process, the shortest length of our learned rules is 2 (i.e., the length of a rule is defined as the number of predicates in the rule body), we are unable to learn identical rules in the form of P1->P1 (e.g., sister(A,B) -> sister(A,B)) as its length is less than 2. As no identical rules are presented in the rules, there is no direct data leakage even if the test triple to be predicted is in the KG.
> > >
> > > That being said, to avoid an unfair comparison due to the additional observations in KG, which may give our method an advantage (e.g., if a test triple t1 is involved in a rule (e.g., t1^t3->t2) that can derive another test triple t2), we follow your suggestion to provide new results under the fair comparison by removing all the test triples. The results of the three small datasets are shown below. We can see that our conclusion is still the same, though the performance of our approach decreased a little bit. We are collecting the results on the larger KGs and we will post them here soon.
> > >
> > > $$
> > > \begin{array}{@{}|c|ccc|ccc|ccc|@{}}\hline
> > > & &\text{Family}& & &\text{Kinship}& & &\text{UMLS} & \\\\
> > > &\text{MRR} & \text{Hit@1} & \text{Hit@10} & \text{MRR} & \text{Hit@1} & \text{Hit@10} & \text{MRR} & \text{Hit@1} &  \text{Hit@10} \\\\\hline
> > > \text{TransE} & 0.45 & 22.1 & 87.4 & 0.31 & 0.9 & 84.1 & 0.69 & 52.3 & 89.7 \\\\\hline
> > > \text{DistMult} & 0.54 & 36.0 & 88.5 & 0.35 & 18.9 & 75.5 & 0.391 & 25.6 & 66.9 \\\\\hline
> > > \text{ComplEx} & 0.81 & 72.7 & 94.6 & 0.42 & 24.2 & 81.2 &  0.41 & 27.3 & 70.0 \\\\\hline
> > > \text{RotatE} &  0.86 & 78.7 & 93.3 & \textcolor{red}{\textbf{0.65}} & \textcolor{red}{\textbf{50.4}} & \textcolor{red}{\textbf{93.2}} & 0.74 & 63.6 & 93.9 \\\\\hline
> > > \text{Neural-LP} & 0.88 & 80.1  & 98.5  &  0.30 & 16.7 & 59.6 & 0.48 & 33.2 & 77.5 \\\\\hline
> > > \text{DRUM} & 0.89 & 82.6 & 99.2 & 0.33 & 18.2 & 67.5 & 0.55 & 35.8 & 85.4 \\\\\hline
> > > \text{RNNLogic} & 0.86 & 79.2 & 95.7 & \textcolor{blue}{\textbf{0.64}} & \textcolor{blue}{\textbf{49.5}} & 92.4 & 0.75 & 63.0 & 92.4 \\\\\hline
> > > \text{RLogic} & 0.88 & 81.3 & 97.2 & 0.58 & 43.4 & 87.2 & 0.71 & 56.6 & 93.2 \\\\\hline
> > > \text{NCRL\\_test\\_excluded}&\textcolor{red}{\textbf{0.91}} & \textcolor{red}{\textbf{85.2}} & \textcolor{red}{\textbf{99.3}} & \textcolor{blue}{\textbf{0.64}} & 49.0 & \textcolor{blue}{\textbf{92.9}} & \textcolor{red}{\textbf{0.78}} & \textcolor{red}{\textbf{65.9}} & \textcolor{red}{\textbf{95.1}}
> > > \\\\\hline
> > > \text{NCRL\\_test\\_included} & 0.92&85.6 & 99.6 & 0.65 & 49.4 &  93.6 & 0.78 & 66.1 & 95.2\\\\\hline
> > > \end{array}
> > > $$
> > >
> > > (2) Confirmation of the baseline results.
> > >
> > > The reported results of NeuralLP and DRUM are taken directly from RNNLogic [1]. The lower results of DRUM might be caused by the different split used by RNNLogic.
> > >
> > > [1] Qu M, Chen J, Xhonneux L P, et al. Rnnlogic: Learning logic rules for reasoning on knowledge graphs[J]. arXiv preprint arXiv:2010.04029, 2020.

---

> > > > ### Author Response · Authors · 2022-12-10
> > > > **Follow-up Experiment Results**
> > > >
> > > > We thank the reviewer for the valuable comments, which are very helpful for us to improve our work. We follow your suggestion to provide new results for the three large datasets, which are shown below. We can see that our conclusion is still the same for large KGs.
> > > >
> > > > $$
> > > > \begin{array}{@{}|c|ccc|ccc|ccc|@{}}\hline
> > > > & &\text{WN18RR}& & &\text{FB15K-237} & & &\text{YAGO3-10}& \\\\
> > > > &\text{MRR} & \text{Hit@1} & \text{Hit@10} & \text{MRR} & \text{Hit@1} & \text{Hit@10} & \text{MRR} & \text{Hit@1} &  \text{Hit@10} \\\\\hline
> > > > \text{TransE} & 0.23 & 2.2 & 52.4 & 0.29 & 18.9 & 46.5  & 0.36 & 25.1 & 58.0  \\\\\hline
> > > > \text{DistMult} & 0.42 & 38.2 & 50.7 & 0.22 & 13.6 & 38.8  & 0.34 & 24.3 & 53.3\\\\\hline
> > > > \text{ConvE} & 0.43 & 40.1 & 52.5 &\textcolor{red}{\textbf{0.32}} & 21.6 & 50.1 & 0.44 & 35.5 & 61.6\\\\\hline
> > > > \text{ComplEx} & 0.44 & 41.0 & 51.2 & 0.24 & 15.8 & 42.8 & 0.34 & 24.8 & 54.9\\\\\hline
> > > > \text{RotatE} & 0.47 & 42.9 & 55.7 & \textcolor{red}{\textbf{0.32}} & \textcolor{red}{\textbf{22.8}} & \textcolor{red}{\textbf{52.1}}  & \textcolor{red}{\textbf{0.49}} &  \textcolor{red}{\textbf{40.2}} & \textcolor{red}{\textbf{67.0}}\\\\\hline
> > > > \text{Neural-LP} & 0.38 & 36.8 & 40.8 & 0.24 & 17.3 & 36.2 & - & - & - \\\\\hline
> > > > \text{DRUM} & 0.38 & 36.9 & 41.0 & 0.23 & 17.4 & 36.4  & - & - & - \\\\\hline
> > > > \text{RNNLogic} & 0.46 & 41.4 & 53.1 & 0.29 & 20.8 & 44.5  & - & - & - \\\\\hline
> > > > \text{RLogic}  & 0.47 & 44.3 & 53.7 & \textcolor{blue}{\textbf{0.31}} & 20.3 & \textcolor{blue}{\textbf{50.1}} & 0.36 & 25.2 & 50.4 \\\\\hline
> > > > \text{NCRL\\_test\\_excluded}& \textcolor{red}{\textbf{0.67}} & \textcolor{red}{\textbf{56.3}}  & \textcolor{red}{\textbf{85.0}} & 0.30 & \textcolor{blue}{\textbf{20.9}} & 47.3  & \textcolor{blue}{\textbf{0.38}} & \textcolor{blue}{\textbf{27.4}} & \textcolor{blue}{\textbf{53.6}}\\\\\hline
> > > > \text{NCRL\\_test\\_included} & 0.67 & 56.8  & 85.2 & 0.31 & 22.0 & 48.2  & 0.38 & 27.8 & 54.1\\\\\hline
> > > > \end{array}
> > > > $$

---

> > > > > ### Comment · Reviewer_XZA5 · 2022-12-13
> > > > > **Comments**
> > > > >
> > > > > Thanks for the updates. I think the clarification makes sense to me. I'm a bit disappointed that the authors should have tried fixing the issues right after the public comment came out, rather than waiting until the reviewers pointed it out, as the previous comparison is clearly flawed. Nevertheless, since the provided results make sense to me, I will still recommend accepting the paper (though no longer a strong accept). In addition, I strongly suggest the authors carefully inspect the implementation to make sure the comparison is fair this time, and also include the source code for reproducing the full result table above in the final version.

---

### Public Comment · ~Zhaocheng_Zhu1 · 2022-11-07
**Is there data leakage in the evaluation?**

I quite appreciate the novel idea of this paper. However, it looks like there is a data leakage problem in the evaluation.

In the supplementary material `code/kg_completion_fast.py`, line 186 creates `r2mat` by merging facts, train, validation and test triplets. This matrix is used as the grounding for rule inference in `RuleDataset`. To my understanding, if the model learns some identical rules that copy any rule body to predict its rule head, the model will exploit this data leakage and can easily achieve perfect prediction.

Besides, I only found the fact triplets are flipped in the code. The standard KGC evaluation requires to predict both $(h, r, ?)$ and $(?, r, t)$, which is used by all the baselines in this paper. Only evaluating $(h, r, ?)$ usually results in higher performance. Can the author confirm if they have flipped the training / validation / test triplets?

---

> ### Author Response · Authors · 2022-11-07
> **Reply for data leakage in the evaluation**
>
> Thank you for the comment on our work.
>
> First of all, we would like to clarify that there is no data leakage in the evaluation. We follow the same setting as NTPs for the evaluation. The basic idea is that given a triple (h,r,t), we can leverage all the triples besides (h,r,t) as the background knowledge to infer the target triple (h,r,t). The matrix r2mat is used as the background knowledge when instantiating the learned rule to infer new facts. Therefore, the matrix r2mat can merge facts, train, validation, and test triplets. We agree that when the learned rules include identical rules, such as r1(a,b)<-r1(a,b), the model can exploit data leakage easily. However, we only include the rules whose lengths are over 2 for evaluation, which ensures there is no data leakage.
>
> Second, we follow the setting of NeuralLP to predict both (h,r,?) and (?, r, t). For each relation r, we incorporate its inverse relations r^-1 by preprocessing the KGs to add inverse triples (t, r^-1, h). In this way, (?,r, t) can be evaluated by predicting (t, r^-1, ?).

---

### Decision · Program_Chairs · 2023-01-20

**Decision:**

Accept: poster

**Justification For Why Not Higher Score:**

Issues with initial experimental setup

**Justification For Why Not Lower Score:**

Nice technical contribution with eventually satisfying experiments.

**Metareview: Summary, Strengths And Weaknesses:**

Summary: authors propose an end-to-end approach for learning compositional rules
Strength: interesting approach with good empirical performance
Weakness: Design of experimental setup

**Note From Pc:**

if the above contains the word "oral" or "spotlight" please see: "oral" presentation means -> notable-top-5% and "spotlight" means -> notable-top-25%. As stated in our emails, we are disassociating presentation type from AC recommendations

**Summary Of Ac-Reviewer Meeting:**

All reviewers agreed that the experimental setup was problematic and that the paper can only get accepted if the authors re-run their experiments in a fair setup. Indeed, the authors have done so and thus according to discussion with reviewers they raised their scores accordingly.